# Notch and Hedgehog Signaling Unveiled: Crosstalk, Roles, and Breakthroughs in Cancer Stem Cell Research

**DOI:** 10.3390/life15020228

**Published:** 2025-02-04

**Authors:** Sabina Iluta, Madalina Nistor, Sanda Buruiana, Delia Dima

**Affiliations:** 1Department of Hematology, Iuliu Hatieganu University of Medicine and Pharmacy, 400124 Cluj Napoca, Romania; iluta.sabina@yahoo.com; 2Medfuture Research Center for Advanced Medicine, Iuliu Hatieganu University of Medicine and Pharmacy, 400124 Cluj Napoca, Romania; madalina.nistor7@gmail.com; 3Department of Hematology, Nicolae Testemitanu University of Medicine and Pharmacy, MD-2004 Chisinau, Moldova; sanda.buruiana@usmf.md; 4Department of Hematology, Ion Chiricuta Oncology Institute, 400015 Cluj Napoca, Romania

**Keywords:** Notch signaling pathway, Hedgehog signaling, cancer stem cells, tumor biology, cancer therapy, tumor microenvironment

## Abstract

The development of therapies that target cancer stem cells (CSCs) and bulk tumors is both crucial and urgent. Several signaling pathways, like Notch and Hedgehog (Hh), have been strongly associated with CSC stemness maintenance and metastasis. However, the extensive crosstalk present between these two signaling networks complicates the development of long-term therapies that also minimize adverse effects on healthy tissues and are not overcome by therapy resistance from CSCs. The present work aims to overview the roles of Notch and Hh in cancer outburst and the intersection of the two pathways with one another, as well as with other networks, such as Wnt/β-catenin, TGF, and JAK/STAT3, and to explore the shaping of the tumor microenvironment (TME) with specific influence on CSC development and maintenance.

## 1. Introduction

The Notch and Hedgehog (Hh) pathways are two of the most relevant regulators of cell differentiation, tissue patterning, and maintenance [1,2]. Highly conserved, their structural roles have long been established as controlling the body architecture during embryogenesis and tissue homeostasis throughout adulthood. In healthy conditions, Notch and Hh affect a diversity of cellular processes, such as proliferation, differentiation, and apoptosis, and are often controlling each other [3]. Notch signaling is crucial in the maturation of several tissues, like bone, muscle, heart, vasculature, liver, stomach, and the hematopoietic system. Specifically, Notch is crucial in mesenchymal progenitor cell (MPC) growth and formation by regulating the production of bone cells through specific receptors and ligands (NOTCH1, NOTCH2, Dll1, and JAG1) [4]. Moreover, it also has a spatial and temporary input into bone fraction healing, with MPC depletion and fracture nonunion in the absence of Notch signaling, as observed in mouse models [5]. During the cardiomyogenesis process, Notch signaling has an impactful contribution in promoting the maturation of the atrioventricular architecture with the formation of the heart valve and the development of the epithelial–mesenchymal transition (EMT) by downregulating VEGFR2, the main negative regulator of EMT in the atrioventricular chambers [6,7]. In addition, it was noticed that embryos lacking Notch1, Rbpj, Hey1/Heyl, or Hey2 proteins could not develop EMT, which leads to an invasion of endocardium cells in the cardiac jelly [8]. Subsequently, Notch signaling is also pivotal in vascular endothelial cell development. Several studies have shown that deficiencies in Notch-specific molecules, like Delta-like 4 (DLL4), NOTCH2, NOTCH3, and NOTCH4, impact normal development severely, with vascular dysplasia, aortic defects, and abnormal development of multiple organs [9,10]. Notch signaling was also observed to promote biliary duct cell differentiation, as well as the specific timing of the process, with Sox9 as a key downstream target [11,12]. NOTCH2 is also indispensable in liver regeneration processes, like differentiation of the progenitor cells of the liver tissue [13,14]. In the gastrointestinal tract, expression of Hes1 through Notch control guides the intestinal stem cells into epithelial cell differentiation [15]. Additionally, NOTCH receptors are involved in maintaining the antral cell stem population of the stomach, while the DLL 1 and 4 and JAG1 ligands regulate the differentiation of pancreatic progenitors of the α cells [16]. Lastly, the development of lymphocytes, macrophages, granulocytes, and dendritic cells and the maintenance of the stemness of hematopoietic stem cells are all Notch-dependent processes [17,18,19,20,21].

Hh signaling, on the other hand, drives pivotal downstream regulations of transcription factors that control the morphogenesis and spatial patterning of the organism, like the differentiation and mineralization of bone cells [22], activation of myosatocytes through Myf5 and MyoD expression for skeletal muscle repair [23], control of hair follicle stem cells and epidermal homeostasis [24], intestinal stem cell differentiation and proliferation [25], and liver regeneration, via EMT interaction, mediated by Hh [26].

Although distinct in their roles, the two pathways usually exhibit a bidirectional regulation, with frequent overlapping and crosstalk. For example, in the developmental stages of the spinal cord, the expression of the homeodomain expression factor (HD) and proneural basic helix-loop-helix proteins (bHLH) is critically controlled by the duration and level of the Hh signaling pathway, with indirect downstream effects on the Notch control and Notch-associated jagged gene expression and Fringe proteins [27,28]. Hh can also directly determine neuronal differentiation by activating the promoters of *Hes1*, a classic target gene of Notch, via Gli protein control [29]. On the other hand, Notch signaling helps maintain Hh responsiveness by maintaining the stability of essential proteins, like Gli transcription factors, and by keeping the progenitor cells receptive to Hh signaling [30,31], thus guiding the cell fate to specification in the central nervous system development.

Both pathways are closely regulated for the correct development of the embryo, and their improper regulation during this stage may lead to a predisposition for cancer onset by disrupting homeostasis and stem cell control [32]. Recent studies have shifted the focus from their roles in development to their contributions to cancer biology. Dysregulation of Notch and Hh signaling has been implicated in tumorigenesis, where abnormal activation or inhibition drives the progression of several tumors [3]. First, the defective signaling of Notch and Hh during embryogenesis can lead to an unbalanced population of stem cells and differentiated cells in several tissues, with later consequences into CSC development and maintenance of population through self-renewal, which then contributes to processes like neoplasia, metastasis, drug resistance, and high recurrence rate [3]. The survival of Notch–Hh-positive CSCs and their ability to evade the immune system has also been associated with the hypoxic tumor microenvironment (TME), which was proved to maintain the population of undifferentiated stem cells and also to reprogram the non-stem cancer cells into CSCs, with activating Notch-target genes [33]. Second, Notch and Hh defects can lead to tissue-specific cancer. For instance, mutations of Ptch and Smo, Hh components, can lead to the Gorlin syndrome (GS), which manifests as basal cell carcinoma [34]. Additionally, Notch-associated CSCs have been identified in the early stages of a large number of tumors, like breast [35], colorectal [36], pancreatic [37], brain [38], lung [39], and skin cancers [40], as well as hematological malignancies [41], exercising roles like tumor initiation, EMT, progression, and hypoxia maintenance. Lastly, Notch-Hh associated developmental defects can have a long-term impact on the microenvironment, with morphological and physiological shaping that favors chronic inflammation, CSCs maintenance, and tissue remodeling, setting the stage for prospective cancer development [42,43].

Notch and Hh signaling do not operate in isolation but have complex interplays with other signaling pathways, which exhibit significant roles in CSCs and cancer progression. Most particularly, the crosstalk between Notch and Wnt/β-catenin has an important influence in determining the balance between the stem cells and the specialized ones. The activation of Notch further activates Wnt to promote the proliferation of CSCs [44]. Additionally, Hh interacts with the PI3K/Akt pathway in pathological conditions. The defective activation of Hh enhances the Akt signaling, thus facilitating the survival and growth of CSCs [45]. Furthermore, both Notch and Hh exhibit significant crosstalk with the transforming growth factor-beta (TGF-β) in promoting EMT, together with Wnt/β-catenin, which is a crucial process involved in cancer metastasis and CSC maintenance [46].

The present article aims to explore the roles of Notch and Hh signaling in normal tissue development, highlight the aberrations encountered in the pathways that contribute to tumorigenesis, and focus on cancer stem cell (CSC) origin and maintenance within the TME. Additionally, it provides current insights into targeted cancer therapies and attempts to overcome treatment resistance.

## 2. Structure and Function of Notch Signaling

The activation of Notch signaling is a pathway that drives carcinogenesis and is also highly conserved, with crucial roles in determining cell fates, proliferation, differentiation, and apoptosis [47].

A particular feature of the Notch signaling cascade is the requirement for direct contact between adjacent cells, as both the ligands and receptors are found on the plasma membranes of the interacting cells (Figure 1). The receiving cells usually experience changes in cell fate due to the Notch signals. Such interactions are vital to processes such as hematopoiesis, immune regulation, neural stem cell survival, maturation of the colorectal epithelial, and breast development [48,49,50].

Structurally, four Notch receptors have been identified, each an extracellular heterodimer and a transmembrane domain held together by hydrogen bonds [51]. The extracellular segment contains a sequence of multiple epidermal growth factor-like domains, which mediate the interactions with Notch receptor–ligands [52]. The intracellular fragment of the receptor serves as a transcription factor. Notch receptors have five known ligands: Jagged-, Jagged-2, and Delta-like ligands (DLL1, DLL3, and DLL4), localized on the receptor-expressing neighboring cells [53].

Upon receptor–ligand binding, these ligands trigger an initial cleavage event at a particular site on the Notch receptor. This site is subsequently cleaved by the ADAM MMPs [54], releasing the ligand-bound Notch fragment into the extracellular space, where it undergoes endocytosis and degradation by the adjacent ligand-expressing cell. Another cleavage by the γ-secretase complex releases the Notch intracellular domain (NICD) [8], which then undergoes nuclear translocation [55,56]. In the absence of NICD, the CBF-1-Suppressor of Hairless/Lag1 (CSL-1) interacts with co-repressors, which block their transcriptional activity [57,58,59]. NICD binds to CSL-1 as a transcriptional co-activator, displacing co-repressors and enabling the binding of Mastermind-like proteins (MAML) and others, forming the Notch transcription complex. This complex activates the transcription of effector proteins, like the Hairy/Enhancer of Split (HES) family, the Hey family, and factors like c-Myc and cyclin D1 [59,60], which promote cell proliferation and other cell-specific functions.

Since developmental pathways such as Notch are critical for normal stem cell maintenance, they often dysregulate in cancer. Altered Notch signaling can hinder the normal differentiation of stem cells, contributing to the development of malignancies like T-cell acute lymphoblastic leukemia (T-ALL), glioblastoma multiforme, and cancers of the colon, pancreas, breast, ovary, and prostate [61,62,63,64,65,66,67,68]. Genetic mutations also disrupt normal Notch signaling. Activating mutations in Notch receptors are found in about 60% of T-ALLs, where the receptor becomes active independent of ligand binding or its degradation is delayed due to mutations in the C-terminal PEST domain [69], which usually marks NICD for degradation by the proteasome. In addition, around 10% of chronic lymphocytic and myeloid leukemias, as well as some breast adenocarcinomas, exhibit Notch mutations, which are associated with poorer survival rates [70]. In contrast, loss-of-function mutations in Notch occur in cancers like skin cancer, medullary thyroid cancer, cervical cancer, small cell lung cancer, and hepatocellular cancer (HCC). Notch may act as a tumor suppressor in these cases [71,72].

Other mechanisms that disrupt Notch signaling can contribute to cancer development. For example, Notch hyperactivity may arise from mutations in Numb, a Notch repressor, which is lost in about half the cases of breast cancer [73]. Additionally, the RNA-binding protein Musashi-1, which plays a role in translational inhibition, can cause Numb inactivation and Notch-1 overexpression. Reducing Musashi-1 levels decreases Notch-1 signaling, leading to increased apoptosis and suppressed tumor growth, possibly due to the role of Musashi-1 in maintaining stem-like properties in cancer [74,75,76].

## 3. CSC and Notch Inhibition

The significance of Notch signaling extends beyond its developmental role in normal stem cells to include its function in CSCs [77]. Elevated Notch activity and increased expression of Notch family genes in cancer stem suggest that Notch may drive rapid tumor growth and aggressive behavior [78]. This has led to the hypothesis that inhibiting the Notch pathway could be a promising therapeutic strategy in cancer treatment. Supporting this idea, Fan et al. demonstrated that blocking Notch signaling with γ-secretase inhibitors reduced the development of glioblastoma neurospheres in in vitro studies, decreased the expression of CSCs markers (CD133, NESTIN, BMI1, and OLIG2), and prevented tumor formation in mice injected with glioblastoma cells treated with γ-secretase inhibitors. This treatment also significantly prolonged the survival of mice with intracranial xenografts [79]. Another study using murine orthotopic xenografts derived from temozolomide-sensitive and resistant glioblastoma neurosphere lines found similar results: Notch inhibition led to tumor growth suppression, decreased stem cell marker expression, and induced glial differentiation. These effects persisted even after therapy concluded [80], highlighting Notch’s potential role in maintaining the CSC population.

The involvement of Notch in the EMT process, which is crucial for both embryonic development and cancer progression, may partially explain its role in stem cell renewal and maintenance [77]. EMT is characterized by a visible change in cell morphology from an epithelial to a fibroblast-like appearance, with associated loss of cell adhesion, cell polarity, and increased migration and invasiveness [81]. Significant CSC-like subpopulations are also present in the matrix of such modified cells [82,83,84,85]. Notch signaling promotes EMT by downregulating E-cadherin expression through the activation of Slug, thus initiating the β-catenin pathway, as observed in breast cancer cells [86]. Furthermore, TGF-β-induced EMT can be stopped by inhibiting Notch-1 signaling [87].

γ-secretase inhibitors act effectively against cells with high Notch activity, such as CSCs, but combining them with chemotherapy in order to target bulk tumor mass, which consists of non-stem CSCs, would increase the effectiveness of the anti-cancer strategy. This approach was tested in pancreatic cancer xenografts using a selective γ-secretase inhibitor, combined with gemcitabine, a first-line treatment for pancreatic ductal adenocarcinoma. While gemcitabine alone did not induce tumor regression or deplete pancreatic CSCs, the combination therapy led to significant tumor regression in three out of four xenografts, reduced CSCs, and had a more sustained effect without further treatment compared to gemcitabine alone. This mechanism promoted apoptosis, reduced cell proliferation, inhibited angiogenesis, and controlled metastatic advancement [88]. These findings are particularly relevant, as gemcitabine-resistant pancreatic cancer cells often exhibit increased Notch activity, which promotes EMT, stem cell marker expression, and chemotherapy resistance [89,90].

Targeting both the stem cell population and the bulk tumor cells is critical in advanced stages of cancer. Initial debulking with conventional chemotherapy maintains organ function, while targeting CSCs with selective inhibitors can help prevent relapse [91]. In breast cancer, a combination of the γ-secretase inhibitor (GSI) MK-0752 with docetaxel (a platinum-based chemotherapy) reduced tumor size more effectively than either agent alone and delayed tumor evolution. Notch inhibition by MK-0752 reduced the breast CSC population and enhanced the efficacy of docetaxel. A phase I clinical trial of this combination in 30 patients with advanced breast cancer showed a reduction in CD44(+)/CD24(-) and ALDH(+) stem cells and their mammosphere-forming capacity [92].

However, the potential adverse effects of GSIs should be considered for long-term use. Notch signaling regulates the balance of intestinal cell types, and GSI-mediated inhibition can cause severe goblet cell metaplasia, leading to diarrhea [93]. Additionally, the γ-secretase complex is involved in amyloid protein production in Alzheimer’s disease, as well as an increased incidence of skin cancers, possibly due to Notch’s tumor-suppressing role in epithelial tissues [69].

Beyond GSIs, other classes of Notch inhibitors are being developed, including monoclonal antibodies targeting Notch ligands and receptors, receptor-like decoys that bind and inactivate ligands, targeted peptides, natural compounds, and agents targeting proteins downstream of the Notch receptor [77]. Some phytochemicals, such as curcumin, sulforaphane, resveratrol, genistein, epigallocatechin gallate, and psoralidin, have been found to inhibit the NICD, preventing its nuclear translocation. These compounds can inhibit cancer growth, induce apoptosis, and reduce migration and invasion, affecting both cancer cells and CSCs [94,95].

As Notch-targeted therapies continue to diversify, advances in understanding the Notch signaling pathway may allow for more personalized cancer treatments. A genomic analysis of the Notch pathway in 16 glioma tumor-initiating cell populations identified genes such as NOTCH1, NOTCH3, MAML1, JAG2, HES1, and DLL-3, which were associated with active Notch signaling and enriched in tumors responsive to treatment. This suggests that GSIs may be particularly effective in glioblastomas with high Notch pathway activation [96]. Rizzo et al. have proposed that for personalized treatment, it is essential to determine not only the expression of specific Notch components and ligands but also whether targeting the entire Notch system or specific components is more effective for each tumor type. The study also emphasizes the need to assess the interaction between Notch and other signaling pathways to design rational combination therapies [59].

## 4. Notch Signaling and the TME

Although details about the various microenvironments supporting cancer stem-like cells in different cancer types are still limited, the occurrence and spread of cancer are heavily connected to the TME. Tumor-associated fibroblasts, endothelial cells, adipocytes, and certain immune cells are key components of this background, playing a significant role in regulating both normal and malignant stem cells through paracrine, juxtacrine, or other short-range signaling methods [97].

### 4.1. Endothelial Cells

Endothelial cells are crucial elements of the TME, especially around CSCs, creating what is known as the “vascular niche”. Studies have shown that glioblastoma stem cells are protected by this environment, which hosts endothelial cells [98]. Notch expression strongly influences both glioblastoma stem cells and tumor angiogenesis. In one study, Hovinga et al. proposed that Notch inhibition could significantly affect the behavior of endothelial cells within the tumoral niche, as well as the glioblastoma cells. On a three-dimensional glioblastoma explant model that mimics the tumor’s cytoarchitecture and stroma, Notch inhibition through GSIs led to a loss in the number of endothelial cells and also to the downregulation of Notch target genes (HES1 and HES5) and also eliminated the expression of the endothelial marker CD105. The important implication observed on tumor angiogenesis and the tumor niche, which consequently led to decreased neurosphere formation and a lower number of CD133(+) glioblastoma cells, highlighted the major function of endothelial cells in Notch signaling within glioblastoma. Additionally, by treating the tumors with a Notch inhibitor for five days prior to irradiation, the proliferation and self-renewal capacity of the tumor explants showed a significant reduction, which proves that this strategy works more efficiently than irradiation alone [99].

Lu et al. also managed to bring additional support for the vital role of the TME by studying its behavior in colorectal cancer. They cultured colorectal CSCs in a conditioned medium derived from freshly isolated endothelial cells and observed that the latter cells managed to enhance the stem-like phenotype of the cancer cells in a significant manner: Aldefluor-positive colorectal cancer cell population by 16-fold, the CD133 positive cells by 7-fold, and the sphere-forming capacity by over 6-fold. This mechanism was due to the paracrine capacity of endothelial cells to induce dedifferentiation on non-stem colorectal cancer cells by secreting Jagged-1. On top of that, the soluble Jagged-1 was depleted in the endothelial cell-conditioned medium by a type of antibody that recognized the N-terminus of Jagged-1, which reduced colorectal cancer cells’ sphere-forming ability. These findings bring additional support to the hypothesis that endothelial cells in the tumor niche promote the CSC phenotype by the enhancement of Notch signaling, independent of cell-to-cell contact [100].

### 4.2. Angiogenesis

Notch signaling is also involved in regulating cancer angiogenesis, with DDL4 as a key factor in suppressing tumor angiogenesis and its known high expression in tumor blood vessels. Angiogenesis is induced by two types of endothelial cells: first, the “Tip” VEGF-stimulated endothelial cells, which lead the migration during angiogenesis when an angiogenic stimulus occurs. VEGF is also involved in DLL-4 expression in Tip cells, which generates the activation of Notch signaling in the neighboring “Stalk” cells, preventing additional Tip cells from forming and ensuring a proper vascular structure. Inhibition of DLL-4 signaling induces uncontrollable vessel sprouting and creates a dense but nonfunctional angiogenic network, which reduces tumor growth and increases tumor hypoxia [101].

Endothelial cells may also originate from CSCs through trans-differentiation or vascular mimicry. First observed in melanoma cases, this phenomenon has been described in other types of cancer [102,103]. For example, glioma CSCs can undergo vascular mimicry under hypoxic conditions [104], and it was also observed that the vast majority of tumor endothelial cells can originate from these CSCs. Although studies proposed that Notch may influence the differentiation process, much still remains unknown [102].

### 4.3. Adipose-Derived Stem Cells (ADSCs)

Similarly to endothelial cells from the TME, ADSCs also have a key role in facilitating tumor growth and proliferation. In one study, it was observed that by co-culturing ADSCs with H358 lung cancer cells, EMT-like alterations occur in the latter cells. Notably, these EMT-like chances were blocked by GSIs, suggesting a link to Notch signaling [105].

### 4.4. Hypoxia

The TME is often hypoxic, especially in the tumor central areas that are affected by poor vascularization and necrosis and also in the invasive front, due to rapid cancerous cells [106]. Hypoxia has been associated with more aggressive tumors and increased levels of invasiveness and metastasis, likely because it develops CSCs [107] and diminishes their differentiation [108]. Hypoxia-inducible factor (HIF) expression is upregulated by hypoxia and also mediates many effects of low oxygen levels, like promoting angiogenesis, tumor growth and spreading, and EMT [109,110]. HIF1α is upregulated by Notch signaling [111], and the molecular HIF1α and Notch1 increase the stability and activity of Notch proteins [112].

In hypoxic conditions, Notch promotes the CSC phenotype and blocks their differentiation, which creates a more invasive and pro-survival characteristic [113]. However, these effects may, in turn, be blocked by Notch inhibitors [108,114] and may also promote E-cadherin expression and AKT phosphorylation inhibition [114]. At the hypoxic invasive front, Jagged-2 and Notch signaling activation are both elevated. Jagged-2 activation under hypoxia conditions induces EMT and maintains cell survival. In the bone marrow stroma, the activation of hypoxia-induced Jagged2 activation enhances CSC growth, which strengthens the importance of the Jagged-2 ligand in the progression of the tumor and metastasis. This is particularly explanatory to why bone tissue is such a frequent metastasis site, since it is known to be hypoxic [77].

## 5. Crosstalk with Other Signaling Pathways

The Notch pathway interacts with several other cancer-related signaling pathways, which is crucial for developing personalized cancer treatments. One such pathway is the PI3K/AKT/mTOR axis, known for regulating cell growth, proliferation, and survival. In many cancers, this pathway is hyperactive. Notch signaling enhances mTOR pathway activity by inducing the transcription of HES-1, a repressor of PTEN, a tumor suppressor that normally inhibits the PI3K/AKT/mTOR pathway. Since PTEN loss is common in cancer, Notch activation can further increase mTOR pathway activity. Blocking Notch signaling with GSIs can therefore reduce activity in both the Notch and mTOR pathways. However, as tumors progress, some cells may become resistant to GSIs due to mutational loss of PTEN, which shifts their “oncogene addiction” to the mTOR pathway, making them sensitive to AKT inhibitors instead [115]. Notch signaling can also upregulate the Insulin growth factor-1 (IGF-1) receptor [116], enhancing mTOR signaling independently of PTEN status. Also, inhibiting mTOR signaling may activate Notch signaling [117], though other studies have shown that AKT inhibition can suppress Notch activity [118]. This complex relationship provides a justification for using combinations of inhibitors targeting both Notch and mTOR pathways [115]. They share downstream effectors like Myc, which Notch promotes and which, in turn, activates the mTOR pathway [119], leading to increased survival of cancer cells by restricting p53-mediated apoptosis. Inhibiting Notch signaling can reduce cancer cell aggressiveness by affecting other pathways, such as mTOR and NF-κB.

The NF-κB pathway is another key regulator of cell survival, differentiation, and proliferation, and it controls the transcription of genes involved in immunity, inflammation, and cancer [120]. Notch and NF-κB signaling can interact at several points in the cascade. First, there is reciprocal transcriptional regulation: Notch signaling can promote the transcription of components of the NF-κB complex [121] and upregulate IκBα, an inhibitor of the NF-κB pathway, at the same time [122]. NF-κB, in turn, can promote the expression of Notch target genes, including members of the HES family [123], and activate Notch in a paracrine manner [124] by increasing the expression of Notch ligands like Jagged1 [125]. Additionally, NICD, the active form of Notch1, can bind directly to the NF-κB complex. Initially thought to inhibit NF-κB activity [126], recent studies suggest that NICD binding may actually prevent NF-κB from being exported out of the nucleus, thereby enhancing NF-κB-mediated gene transcription [127].

The RAS/RAF/MAPK signaling system transmits signals from tyrosine-kinase receptors like EGFR to regulate cell cycle progression, survival, and apoptosis. Notch and RAS networks often crosstalk in cancers such as pancreatic, colorectal, breast, and gliomas [128,129,130,131]. RAS signaling enhances the Notch pathway by upregulating the levels of the Notch ligand DLL-1 and elevating the expression and activity of NICD. Ras also increases presenilin-1, a component of the γ-secretase complex [132]. This crosstalk is fundamental in CSC behavior. In glioblastoma, cooperation between the Notch and RAS pathways leads to the development of CSCs-like populations that express higher levels of nestin, a glioblastoma stem cell marker [133].

Other signaling routes related to cancer may also interact with Notch. In colorectal cancer, Wnt-mediated β-catenin signaling upregulates the Notch ligand Jagged-1 [134] and increases Notch-2 expression [135]. However, in glioblastoma stem-like CD133(+) cells, the combination of Wnt activation and hypoxia-induced HIF-α activity induces neuronal differentiation and suppresses Notch signaling [136], reducing the stem-like population. This suppression may be linked to Wnt-mediated induction of Numb, which inhibits NICD activity [137]. Crosstalk also occurs between Notch and the Hh signaling network, which regulates embryonic development, stem cell maintenance, and cancer progression. The Hh pathway can induce Notch activity by upregulating the Jagged-2 ligand in cancer [138]. In breast CSCs, Notch, Hh, and Wnt together regulate stem cell proliferation and differentiation [139]. There are also direct interactions between Notch and the JAK/STAT signaling system and an antagonistic relationship between Notch and Her2/Neu expression and estrogen receptor (ER) status, which may be clinically significant for therapy [79,140].

Given these interactions between Notch and other cancer-related pathways, it is of great importance to study the combination therapies targeting Notch alongside PI3K/AKT/mTOR inhibitors, NF-κB deterrence, Her2/Neu targets, platinum-based chemotherapies, EGFR inhibitors, and Hh pathway suppressors [50].

## 6. Role of Hedgehog Signaling Pathway in CSCs

The Hh pathway, like the Notch and Wnt signaling pathways, is a highly conserved system that regulates cell proliferation and differentiation, especially during embryonic development. Initially identified in Drosophila, homologous genes have since been found in the mammalian genome, where they have crucial roles in several developmental processes, including cell survival and stem cell maintenance [141], but also in cancer onset.

The Hh pathway is named after its three ligands: Sonic, Indian, and Desert, which induce gene transcription in targeted cells. The signal-receiving cells express Patched receptors (PTCH1 and PTCH2 in vertebrates), which are 12-transmembrane proteins that usually suppress Smoothened (Smo), a receptor resembling G-protein-coupled receptors. Normally, PTCH receptors inhibit Smo activity, reducing the conversion of GLi transcription factors to their active form, leaving them to repress Hh pathway-regulated genes. When Hh ligands bind to PTCH receptors, they relive this suppression, activating Smo and increasing levels of active Gli transcription factors, which then promote the expression of target genes (Figure 2). These genes encode proteins lice c-Myc, cyclin D1, and Patched receptors, some of which participate in a negative feedback loop with Hh signaling [142].

The understanding of the molecular mechanism of Hh signaling led to the formulation of a few hypotheses. For instance, it was suggested that PTCH receptors may transport molecules into Hh-sensitive cells that can further regulate Smo activity [143]. But the precise localization of Hh proteins in specific cellular structures is critical for signaling. In vertebrates, the primary cilium is considered to be the exclusive cell site where Hh signal transduction takes place.

There are still unclear aspects about the molecular mechanisms involved in the regulation of the Hh pathway. After translation, Hh proteins undergo autocleavage and are modified with palmitoyl and cholesteryl groups at their N- and C-terminals [144]. The secretion of processed Hh proteins is regulated by various proteins, including the transmembrane transporter Dispatched, which moves Hh ligands across the secreting cell’s plasma membrane (Figure 1). Glypicans, MMPs, and enzymes involved in heparan sulfate biosynthesis [145] also influence the tissue distribution of Hh ligands [146]. The cholesterol moiety helps determine the form of Hh ligand release, which can vary between monomeric, multimeric, and encapsulated forms depending on auxiliary proteins [147]. Additionally, recent studies suggest a possible involvement of filopodia in Hh transport, though this remains unproven in vertebrates [148].

Several Hh-binding proteins and proteoglycans, like HIP, CDO, BOC, GAS1, and Glypican-3, are directly involved in the regulation of the Hh pathway at the PTCH level [149]. When the Hh ligands are absent, PTCH receptors are located in the primary cilia of vertebrate cells, while Smo proteins are displaced in the membrane of intracellular vesicles [150]. When Hh ligands bind to PTCH, the receptor moves from the cilia and is sequestered in vesicles for degradation, enabling Smo proteins to move to the primary cilium and interact with the Gli-processing system. However, Smo requires additional kinase-mediated phosphorylation to activate Hh signaling [151].

The GPCR-like nature of Smo proteins might also influence pathways such as Gαi, but the exact role in Hh signaling is still unclear. Activated Smo prevents the conversion of Gli proteins into repressors by kinases such as PKA, GSK3β, and CKIα, allowing them to be processed into transcriptional activators through post-translational modifications.

Gli-2 and Gli-3, which are zinc-finger transcription factors, undergo various modifications depending on the Hh signal presence, balancing the production of their activator and repressor forms [142]. In the absence of Hh ligands, CKIα, GSK3β, and PKA kinases phosphorylate Gli proteins. Consequently, βTrCP, an E3-ubiquitin ligase-recruiter, is attached [152], which further triggers the ubiquitination of Gli proteins and their partial degradation by the proteasome complex and the final GliR transcriptional repressors production. SuFu protein also binds to unprocessed Gli, preventing its movement into the nucleus and blocking gene transcription.

When Hh ligands bind to PTCH, Smo is phosphorylated by CKI and GPRK2 [153] and moves to the primary cilium membrane by lateral transport inside the cell membrane [154] or by the fusion between Smo-loaded vesicles with the cell membrane [155]. Smo’s accumulation releases Gli from SuFu [155], prevents Gli phosphorylation by kinases, and omits the formation of GliR, leading to the nuclear translocation of Gli proteins and initiation of target gene transcription.

Furthermore, non-canonical Hh signaling has been described in cancer, involving growth factors like EGF, TGFβ, or PDGFα, which can activate GLi-mediated transcription without Smo [156]. Additionally, a non-GLi-dependent mediated pathway was also associated with cancer development, but its specific involvement is yet to be understood [157].

## 7. Hedgehog Signaling Pathway and Its Role in Cancer Onset

The first evidence of Hh signaling involvement in cancer came from one study showing that the loss of PTCH receptor function leads to nevoid basal cell carcinoma syndrome (Gorlin syndrome) [34]. This autosomal dominant disorder is frequently associated with the outbreak of other neoplasms, like embryonal rhabdomyosarcoma, meningiomas, and odontogenic tumors. PTCH mutations are found in over 90% of sporadic basal cell carcinomas and around 20% of medulloblastomas. Additionally, a ligand-independent alternative of the canonical Hh pathway is formed as a result of loss-of-function mutations in SuFu, activating mutations in Smo, and increased GLi expression. In the ligand-dependent form, Hh ligands originate from tumor cells themselves, creating an autocrine signaling loop seen in certain gastrointestinal, pancreatic, lung, and breast cancers [158]. More commonly, Hh signaling occurs through paracrine interactions between tumor cells and the surrounding environment, activating stromal cells to release growth factors that stimulate mitogenic pathways in nearby cancer cells [159]. The most common examples where paracrine signaling is observed are pancreatic, colon, breast, ovarian, and prostate cancers [158,160]; chronic lymphocytic leukemia [161]; and multiple myeloma [162]. It was noticed that these tumor cells display no cilia protrusions, which might be the starting point of cancer development, since Hh signaling becomes impossible [163]. A reverse paracrine mechanism has also been identified in hematologic malignancies, where stromal cells secrete Hh ligands that enhance tumor cell survival by upregulating Bcl-2, inhibiting apoptosis [161].

Despite these findings, the role of ligand-dependent Hh signaling in cancer remains unclear due to the complex interplay of factors like tumor type, stroma, and crosstalk with other pathways. Hh signaling may serve different roles across cancers: as a tumor driver in basal cell carcinoma, a promoter in small cell lung cancer [164], and a modulator of tumor growth that influences CSCs and residual tumor populations post-treatment.

The Hh pathway also contributes to EMT, a key process in cancer metastasis, by promoting the transcription of proteins like SNAI1, which suppress E-cadherin and enhance cell migration [165]. Disrupting Hh signaling has been shown to inhibit EMT in pancreatic cancer cell lines treated with cyclopamine [166]. Hh signaling can also suppress Wnt signaling, indirectly facilitating EMT by preventing tumor cells from retaining their epithelial state through Wnt’s juxtacrine signals in the TME [167,168].

Apoptosis regulation is another aspect influenced by Hh signaling. Knockdown of Gli-3 via siRNA enhanced the effects of tumor necrosis factor-related apoptosis-inducing ligand (TRAIL) by upregulating the TRAIL receptor and death receptor 4 (DR4) and increasing sensitivity to TRAIL binding [169]. Additionally, cyclopamine-induced inhibition of Smo increased Fas receptor expression and blocked UVB-induced basal cell carcinoma development [170].

Cyclopamine is a terpene alkaloid extracted from wild corn ily (*Veratrum californicum*), with non-toxic effects on non-cancerous cells, as observed in vitro on nh-skp-FB0012 foreskin fibroblasts cells [171], or on human astrocytes [172]. Glioblastoma is another type of cancer with strong resistance to chemotherapy, like common temozolomide and radiotherapy, due to CSCs. Monotherapies and combined therapies of Cyclopamine and/or Temozolomide were administered in vitro on GBM95, GBM02, and GBM03 human glioblastoma cell lines. Alone, it managed to reduce the viability of the cancerous cells, and, combined with the chemotherapeutic, it helped in sensitizing the cells to the medication and inducing apoptosis [171]. Cyclopamine also induced the downregulation of Gli-1 expression in a dose-dependent manner in the EC9706 human esophagus cancer cell line, thus suppressing the Hh signaling pathway [173].

The Gli-1 pathway was also observed to be downregulated as an effect of another natural inhibitor, genistein, of MCF-7 breast cancer cells and also in xenografted naked mice with MCf-7 cells. The level of protein expression of both Gli-1 and Smo was significantly reduced by genistein treatment, as well as the number of breast CSCs [174]. Genistein (4′,5,7-trihydroxyisoflavone) is an isoflavone phytoestrogen present in several soybean foods [175], with known antitumor effects. The exploitation of natural sources that target the Hh signaling pathway has become a hot topic in finding treatments for several types of cancer, as monotherapies and also combined with standard medications, and have shown significant action in inhibiting the stemness of the tumors while having no toxic effect on healthy cells and tissues [176,177]. Despite the promising results reported so far, more studies are required on natural inhibitors, especially on understanding more clearly the mechanism of action in different types of cancer, as well as possible interactions with common therapeuticals, due to their known high pleiotropy and low bioavailability [176].

## 8. Hedgehog Signaling (Hh) and CSCs

Hh signaling has been found to significantly influence the CSC population in various types of cancer, including pancreatic [178], breast [179,180], gliomas [181], gastric [182], and colon cancer [183,184]. In several of these, Hh signaling contributes to aggressive tumor phenotypes that are highly resistant to treatment. For example, in gastric cancer, Song et al. identified a subset of cells expressing CD44, CD24, and CD133, which exhibited higher proliferation rates and resistance to chemotherapy. These cells showed elevated Hh signaling activity [182].

In HCC, Hh signaling has been linked to chemoresistance and invasiveness, largely due to its role in promoting EMT. Treatment with cyclopamine, an Hh pathway inhibitor, reduced the population of CD133+/EpCAM+ HCC CSCs [185]. In pancreatic cancer, combining cyclopamine with gemcitabine resulted in decreased CSCs markers and was associated with tumor regression, highlighting potential therapeutic combinations between Hh inhibitors and standard cancer treatments [186]. Beyond gastrointestinal cancers, similar effects of Hh signaling on CSCs have been observed. In glioma cells, Hh activity induced the expression of genes such as Nanog, which regulate CSC properties [187]. Nanog signaling also appeared to enhance Hh signaling, potentially creating a feedback loop that promotes stemness in glioma CSCs.

In prostate and glioblastoma CSCs, Hh signaling inhibition with erismodegib was shown to prevent EMT, with some of these effects mediated by changes in microRNAs and proteins [188,189]. There is evidence that Hh signaling specifically drives the proliferation of CSCs while having little effect on non-stem cancer cells [190,191].

Hh signaling also contributes to radio- and chemoresistance, primarily driven by CSCs [192,193].

In breast cancer, chemoresistance was found to be associated with P-glycoprotein activity, which correlates with Hh signaling in breast CSCs. Finally, simultaneous inhibition of Notch and Hh signaling sensitized prostate cancer tumor-initiating cells to docetaxel, a chemotherapy drug, by inhibiting AKT and downregulating the anti-apoptotic protein Bcl-2, highlighting the potential of targeting multiple pathways in CSC-driven cancers [194].

## 9. Hh and Crosstalk with Other Signaling Pathways

The Hh signaling pathway interacts with multiple other signaling networks, although the exact mechanism of these interactions remains poorly understood. Some pathways, such as RAS/RAF/MEK/ERK, PI3K/AKT/mTOR, EGFR, and Notch, which are all involved in regulating cell proliferation, growth, and survival, have better-characterized relationships with the Hh axis [195]. In various cancers, such as colon, pancreatic, and lung cancer, K-RAS has been shown to work in conjugation with Hh signaling [196]. In particular, for constitutive K-RAS activation, a hallmark of certain aggressive cancers like pancreatic cancer models, the absence of Smo proteins did not appear to affect Gli-dependent gene expression [197,198]. In melanomas, inhibition of RAS signaling caused Gli proteins to accumulate in the tumor cell cytoplasm, suggesting that K-RAS may play a role in the nuclear translocation of Gli, a process that is not yet fully understood [199].

Crosstalk with the mTOR pathway also occurs at various points and may contribute to cancer development in specific contexts.

The redundancy and interaction between Hh signaling and other pathways have potential therapeutic implications. In glioblastoma, for example, the simultaneous inhibition of both the Notch and Hh pathways significantly increased the toxicity of the chemotherapy drug temozolomide in the CSC population compared to inhibiting either pathway alone. This highlights the importance of targeting multiple pathways to improve treatment efficacy in certain cancers [200].

## 10. Conclusions

Recent advances in our understanding of Hh signaling, both in normal physiological processes and in cancer, have significantly influenced the development of new treatment strategies for various cancer types. 

The Notch signaling pathway is a highly conserved signaling pathway with essential roles in regulating cellular processes like cell fate, proliferation, differentiation, and apoptosis. Since it has such a large implication in these functions, any dysregulation of the pathway can lead to human cancer and also affect cell populations, like the CSCs. As it has already been proven that CSCs dictate the level of aggressiveness of the tumoral cells and their resistance to therapies, targeting the Notch signaling system with molecular strategies could be a very efficient problem-solving strategy in the control and elimination of the CSCs.

Recent advances in our understanding of molecular signaling pathways, both in normal physiological processes and cancer, have significantly influenced the development of new treatment strategies for various cancer types. While there are still unanswered questions regarding the precise molecular mechanisms of Hh signaling and its deregulation in cancer, it is clear that Hh signaling plays a crucial role in oncogenesis. Targeting this pathway holds great promise for enhancing efforts to combat the key characteristics of cancer, including uncontrolled cell proliferation, tumor aggressiveness, and metastasis, all of which are hallmarks of CSCs.

## Figures and Tables

**Figure 1 life-15-00228-f001:**
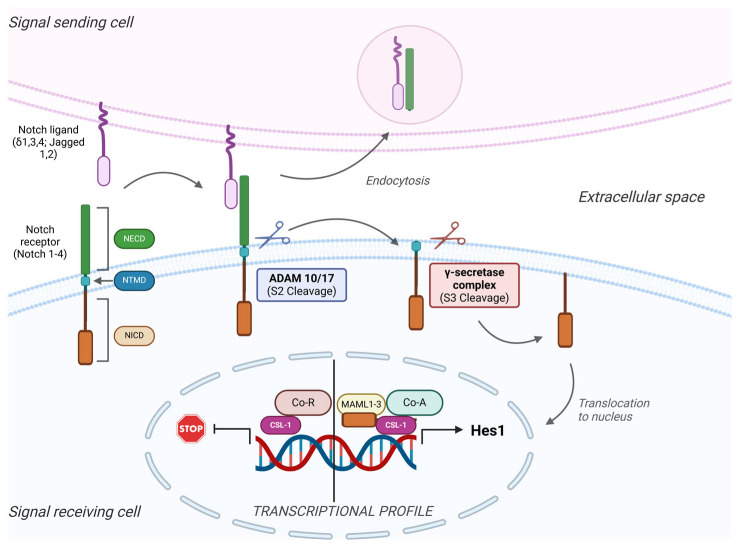
Overview of the general mechanisms involved in Notch signaling. The Notch signaling pathway involves adjacent cells, and their communication involves four receptors (Notch 1–4) and five types of ligands (Delta 1, 3, and 4 and Jagged 1 and 2). When the ligand binds to the extracellular domain of the receptor (NECD), two subsequent cleavages occur. First, the metalloprotease complex ADAM 10/17 cuts the NECD from the rest of the receptor (S2 cleavage), followed by the S3 cleavage mediated by the γ-secretase complex. This proteolysis leads to the detachment of the intracellular domain (NICD) from the transmembrane domain (TMD) and its translocation into the nucleus. Once inside the nucleus (right), it attaches to the CSL, a DNA binding protein, together with other co-activator (Co-A) factors like Mastermind-like proteins (MAML 1–3), and activates the transcription and translation of target genes. On the other hand, when the NICD is not released from the membrane, co-repressors (Co-R) attach to the CSL protein and inhibit the activation of the target genes.

**Figure 2 life-15-00228-f002:**
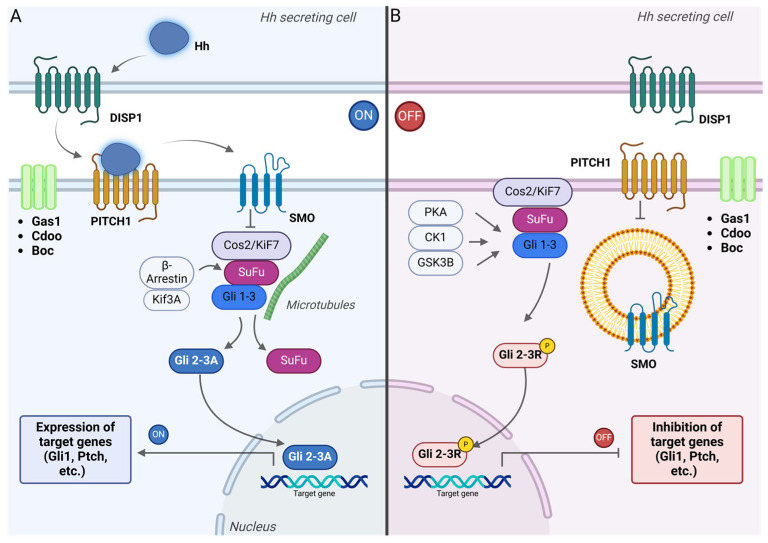
Signal transduction in the Hedgehog (Hh) signaling pathway. (**A**) Hh ligand secretion depends on the transmembrane transporter Dispatched. When Hh ligands bind to PITCH receptors, they inhibit PITCH activity, which allows Smo proteins, which are similar to G-protein-coupled-receptors, to be phosphorylated and transported to the membrane of primary cilia in vertebrates. This prevents the phosphorylation of Gli proteins, promoting their conversion into Gli 2-3A activator form, and moves into the nucleus to activate the expression of Hh-related genes [142]. (**B**) When no Hh signal is secreted, PITCH inhibits the activation of Smo. The Gli 2-3 complex is retained in the cytosol by SuFu and undergoes partial phosphorylation under the influence of a protein kinase complex (PKA, CK1, and GSK3β). The Gli 2-3 repressor form is synthesized (Gli 2-3R) and translocated into the nucleus, where it binds to the DNA and inhibits the expression of the target genes.

## Data Availability

Any inquiries regarding supporting data availability of this study should be directed to the corresponding author.

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
