# Peer review of "Notch and Hedgehog Signaling Unveiled: Crosstalk, Roles, and Breakthroughs in Cancer Stem Cell Research"

_life, 2025, doi:10.3390/life15020228_

Round 1
Reviewer 1 Report
Comments and Suggestions for Authors
The manuscript entitled "Molecular Signaling Pathways in the Cancer Cell" summarizes key molecular signaling pathways related to cancer stem cells (CSCs). I have carefully reviewed the article and would like to highlight the following concerns:
Major
What is the hypothesis driving this review? There are already numerous reviews on this topic in the literature. What is the specific purpose or unique angle of this review? What is its aim?
Minor
1) Title: The title of the manuscript should be revised. It mentions Molecular Signaling Pathways, but the manuscript focuses specifically on the Notch and Hedgehog pathways.
2) Running Title: The running title should be removed.
3) Figure 1: Was this figure generated by the authors, or is it adapted from another review article? Please clarify.
4) Content Enhancement: To enhance the manuscript's quality, consider including additional CSC-associated pathways.
Comments on the Quality of English LanguageEnglish language is fine.
Author Response
The manuscript entitled "Molecular Signaling Pathways in the Cancer Cell" summarizes key molecular signaling pathways related to cancer stem cells (CSCs). I have carefully reviewed the article and would like to highlight the following concerns:
Major:
What is the hypothesis driving this review? There are already numerous reviews on this topic in the literature. What is the specific purpose or unique angle of this review? What is its aim?
Answer: Thank you for the observation. A new section called “Introduction” was added to the manuscript, and the aim of the review was explained at the end of the respective section.
Minor:
1) Title: The title of the manuscript should be revised. It mentions Molecular Signaling Pathways, but the manuscript focuses specifically on the Notch and Hedgehog pathways.
A: Thank you for the valuable observation. The title has been rephrased.
2) Running Title: The running title should be removed.
A: Thank you for the observation. The Running Title was removed.
3) Figure 1: Was this figure generated by the authors, or is it adapted from another review article? Please clarify.
A: Thank you. An original figure was regenerated by the authors in order to correspond to the updated literature.
4) Content Enhancement: To enhance the manuscript's quality, consider including additional CSC-associated pathways.
A: Thank you, the crosstalk with other pathways topic has been adressed in the “Introduction” section.
Reviewer 2 Report
Comments and Suggestions for Authors
I would recommend the authors to be more specific in the title since they are not covering the entire cell signalosome in cancer, instead they overview two specific signaling pathways: Notch and Hedgehog. Also, since there are many other reviews overviewing the same signaling pathways (in cancer and not), it would be beneficial for the review to add clarity on the purposes of this work, e.g. clarifying what`s new in the field, and/or highlighting the most recent findings in the field and critically compare them to well-known studies in the field. On the same line, I would recommend updating information in the figures.
Finally, my main concern is about the (almost) lack of references. This is important for both mentioning milestone articles that made the history of the field, and updating the history with the most recent studies on the field.
The authors mention several times that both Notch and Hedgehog are crucial during developmental stages. Can you add some more details about this? Or mention a fewer times in the text?
This review comprehensively overviews the various molecular players and mechanistic patterns of the Notch and Hedgehog signaling pathways in cancer. Since no other signaling pathways are described, I would recommend the authors to consider being more specific in the title of the review. Also, I would recommend including a small section, which can be in the introduction/conclusion, where the authors could introduce/discuss these two pathways and their crosstalk in cancer so to add clarity to the purpose of this review (why the interest in these two pathways).
Importantly, because of very small bibliography, finding the primary source of the mentioned studies was not possible. It is necessary to cite every single study that was mentioned in this review (at least 1 citation per scientific statement, result, information, this means almost 4x more references in the Bibliography.
Both the signaling pathways are crucial during development. This was mentioned a few times in the manuscript. Do developmental defects involving these pathways impact/influence cancer stem cell development or cancer onset later on in life?
Minor comments:
- Please define each acronym/abbreviation at its first appearance in the text (e.g. EMT..)
- The review would benefit from more figures illustrating a few examples of the mechanisms/molecules reported
This review aims to summarize the role of the Notch and Hedgehog signaling pathways in cancer. The authors provide a detailed description of mechanistic patterns and molecules involved in these signaling and in the importance in the cancer context. I found it quite clear and straightforward, although most of the references are missing.
I would recommend minor changes- including bibliography upgrade- for publication in Life.
Author Response
I would recommend the authors to be more specific in the title since they are not covering the entire cell signalosome in cancer, instead they overview two specific signaling pathways: Notch and Hedgehog. Also, since there are many other reviews overviewing the same signaling pathways (in cancer and not), it would be beneficial for the review to add clarity on the purposes of this work, e.g. clarifying what`s new in the field, and/or highlighting the most recent findings in the field and critically compare them to well-known studies in the field. On the same line, I would recommend updating information in the figures.
Finally, my main concern is about the (almost) lack of references. This is important for both mentioning milestone articles that made the history of the field, and updating the history with the most recent studies on the field.
Answer: Thank you for all your constructive observations. The title has been rephrased in a more specific manner. Also, a new section called “Introduction” was inserted in the manuscript, where we addressed in a clearer manner the aim of the review. Regarding the figure, it was entirely redone in order to correspond to the updated literature. Finally, all references have been inserted in the manuscript.
The authors mention several times that both Notch and Hedgehog are crucial during developmental stages. Can you add some more details about this? Or mention a fewer times in the text?
A: Thank you for the comment. In the “Introduction” part, more details about the involvement of the two pathways in the development of the organism were introduced.
This review comprehensively overviews the various molecular players and mechanistic patterns of the Notch and Hedgehog signaling pathways in cancer. Since no other signaling pathways are described, I would recommend the authors to consider being more specific in the title of the review. Also, I would recommend including a small section, which can be in the introduction/conclusion, where the authors could introduce/discuss these two pathways and their crosstalk in cancer so to add clarity to the purpose of this review (why the interest in these two pathways).
A: Thank you, the title was rephrased in a more specific way. Also, addressing the crosstalk between the two signalling pathways, this topic has been detailed in the “Introduction” section.
Importantly, because of very small bibliography, finding the primary source of the mentioned studies was not possible. It is necessary to cite every single study that was mentioned in this review (at least 1 citation per scientific statement, result, information, this means almost 4x more references in the Bibliography.
A: Thank you for your observation. We have reintroduced all the bibliography cited in the manuscript.
Both the signaling pathways are crucial during development. This was mentioned a few times in the manuscript. Do developmental defects involving these pathways impact/influence cancer stem cell development or cancer onset later on in life?
A: Thank you for the meaningful comment. We have addressed this idea in the “Introduction” section.
Minor comments:
- Please define each acronym/abbreviation at its first appearance in the text (e.g. EMT..)
R: Thank you for your observation. We have revised the entire document and have made the necessary changes.
- The review would benefit from more figures illustrating a few examples of the mechanisms/molecules reported
R: Thank you for the argument. Two original figures were inserted in the manuscript.
This review aims to summarize the role of the Notch and Hedgehog signaling pathways in cancer. The authors provide a detailed description of mechanistic patterns and molecules involved in these signaling and in the importance in the cancer context. I found it quite clear and straightforward, although most of the references are missing.
I would recommend minor changes- including bibliography upgrade- for publication in Life.
R: Thank you for your constructive comment.
Reviewer 3 Report
Comments and Suggestions for Authors
Review comments:
Title: Molecular Signaling Pathways in the Cancer Cell.
The manuscript by Iluta et. al., described a well drafted review about different signaling pathways involved in cancer cells. The authors also discussed about the cross-talk among different pathways that helps in the development of long-term therapies and their clinical implications. Although the manuscript is well curated. I have some comments which needs to be addressed.
Major points:
1. Abstract: The authors have mentioned the text “between three signaling networks”. What are these pathways, please name them? Also, the abstract is very short to get an idea about the context of the review. Please elaborate it.
2. Keywords: write all 3 pathways in keywords.
3. Introduction: The introduction is missing. Please write the introduction in 2-3 paragraphs which describes about the complete review. Then start with the signaling pathways.
4. The authors have included many proteins which functions in the molecular pathway in cancer cells. They authors can also include some other markers/pathways and their recent advancements in p53 mediated apoptosis, GSK3, TNF, hippo signaling, nrf2 pathways etc.
5. The authors are suggested to include figures about the other pathways such as Notch signaling, Wnt signaling etc. to make the review more informative.
Minor points:
1. The point 3 should come after point 4 as it describes about the other pathways.
2. The authors are suggested to revise manuscript for minor grammatical errors.
Author Response
The manuscript by Iluta et. al., described a well drafted review about different signaling pathways involved in cancer cells. The authors also discussed about the cross-talk among different pathways that helps in the development of long-term therapies and their clinical implications. Although the manuscript is well curated. I have some comments which needs to be addressed.
Major points:
- Abstract: The authors have mentioned the text “between three signaling networks”. What are these pathways, please name them? Also, the abstract is very short to get an idea about the context of the review. Please elaborate it.
Answer: Thank you for your observation. The information in the abstract was revised and modified according to your input.
- Keywords: write all 3 pathways in keywords.
A: Thank you, the keywords was selected more carefully specifically linked to the text content.
- Introduction: The introduction is missing. Please write the introduction in 2-3 paragraphs which describes about the complete review. Then start with the signaling pathways.
A: Thank you for the valuable feedback. We have introduced an “Introduction” section.
- The authors have included many proteins which functions in the molecular pathway in cancer cells. They authors can also include some other markers/pathways and their recent advancements in p53 mediated apoptosis, GSK3, TNF, hippo signaling, nrf2 pathways etc.
A: Thank you for the comment. Additional information regarding this topic was introduced in the “Introduction section”.
- The authors are suggested to include figures about the other pathways such as Notch signaling, Wnt signaling etc. to make the review more informative.
A: Thank you for your observation. The manuscript was restructured to be focused on the Notch and the Hedgehog signaling pathways and one figure was inserted for each pathway.
Minor points:
- The point 3 should come after point 4 as it describes about the other pathways.
A: Thank you for the observation. We have applied the requested changes to the final text.
- The authors are suggested to revise manuscript for minor grammatical errors.
A: Thank you. The entire document was revised and corrected.
Reviewer 4 Report
Comments and Suggestions for Authors
This review focuses on the role of the Notch pathway in cancer cells. The authors mainly describe this pathway and its crosstalk with Hedgehog Signalling Pathway and Other Signalling Pathways. The review is comprehensive but the title does not correspond to what it deals with. In my opinion, the role of the Notch pathway in cancer should be mentioned in the title. In addition, for greater clarity the authors should include a picture of the Notch pathway and another that outlines interplay with the other pathways described in the review.
Author Response
This review focuses on the role of the Notch pathway in cancer cells. The authors mainly describe this pathway and its crosstalk with Hedgehog Signalling Pathway and Other Signalling Pathways. The review is comprehensive but the title does not correspond to what it deals with. In my opinion, the role of the Notch pathway in cancer should be mentioned in the title. In addition, for greater clarity the authors should include a picture of the Notch pathway and another that outlines interplay with the other pathways described in the review.
Answer: Thank you for the observation. The title of the manuscript was rephrased in a more specific way, according to your suggestion, and a figure of Notch signaling pathway was inserted in the text.
Round 2
Reviewer 1 Report
Comments and Suggestions for Authors
Dear Authors!
I would like to thank you for taking into account my comments regarding your manuscript.
I have read the introduction section that you added in your manuscript and I think that your review is now suitable for publication.
Reviewer 3 Report
Comments and Suggestions for Authors
The authors have now significantly revised the manuscript with latest information and references. I recommend this review for publication in Life.